# Multi-Perspective Representation to Part-Based Graph for Group Activity Recognition

**DOI:** 10.3390/s22155521

**Published:** 2022-07-24

**Authors:** Lifang Wu, Xianglong Lang, Ye Xiang, Qi Wang, Meng Tian

**Affiliations:** Faculty of Information Technology, Beijing University of Technology, Beijing 100124, China; lfwu@bjut.edu.cn (L.W.); langxianglong@emails.bjut.edu.cn (X.L.); wangqi@emails.bjut.edu.cn (Q.W.); tianmeng@emails.bjut.edu.cn (M.T.)

**Keywords:** group activity recognition, part-based, graph reasoning, video analysis

## Abstract

Group activity recognition that infers the activity of a group of people is a challenging task and has received a great deal of interest in recent years. Different from individual action recognition, group activity recognition needs to model not only the visual cues of individuals but also the relationships between them. The existing approaches inferred relations based on the holistic features of the individual. However, parts of the human body, such as the head, hands, legs, and their relationships, are the critical cues in most group activities. In this paper, we establish the part-based graphs from different viewpoints. The intra-actor part graph is designed to model the spatial relations of different parts for an individual, and the inter-actor part graph is proposed to explore part-level relations among actors, in which visual relation and location relation are both considered. Furthermore, a two-branch framework is utilized to capture the static spatial and dynamic temporal representations simultaneously. On the Volleyball Dataset, our approach obtains a classification accuracy of 94.8%, achieving very competitive performance in comparison with the state of the art. As for the Collective Activity Dataset, our approach improves the accuracy by 0.3% compared with the state-of-the-art results.

## 1. Introduction

Group activity recognition is one of the most challenging tasks in computer vision and attracts many researchers. It has a wide range of practical applications in security surveillance, social role understanding, and sports video analysis. To understand what is happening in a scene involving a group of people, the model not only needs to recognize the individual actions of each person but also needs to explore relations among people [1,2,3].

In the group activity recognition task, it is difficult to model the relationship and to capture the interaction information between individuals in the scene, as we cannot access explicit annotations of the relations between individuals. In order to complete the group activity recognition task and solve the difficulty of modeling interaction information between individuals in a scene, many methods [4,5,6] focused on combining hand-crafted features with graphical models in earlier years. With the development of convolutional neural networks (CNN), due to the excellent performance achieved in image classification tasks [7,8], using convolutional neural networks to solve image-/video-related tasks has become the focus of research in recent years [9,10,11,12,13]. In this situation, many deep learning methods [14,15,16,17,18,19] have been proposed and applied to the group activity recognition task: earlier methods [14,15] utilized convolutional neural networks (CNN) to capture individual appearance features in the scene and combine recurrent neural networks (RNN) to capture the temporal relations of individuals. Recently, attention-based methods [16,17,18,19] modeling spatial-temporal relations between actors have become famous and demonstrated promising results.

The aforementioned methods can generally be summarized in three key steps: (1) extracting the global appearance feature of each individual by a backbone network to a feature vector; (2) exploring the interaction information among people to obtain the relation representation in the context; (3) aggregating individual features as the final group level feature to infer group activity. However, these methods only constructed interactive information directly based on individuals in the scene and ignored the influence of different parts of the body, which may contain more discriminative and robust information for group activity recognition.

In fact, local information from body parts is important for group activity recognition. Some observations can be obtained from Figure 1. In Figure 1a, the group activity of “walking” is significantly related to the legs of most actors, as marked in yellow rectangles. And the features of leg parts and their interaction can bring about more discriminative information for activity prediction. In Figure 1b, the players in the same team are holistically more similar because they wear the same sports suits, and the difference between them is represented by the shapes of legs or hands, as shown in red rectangles. Additionally, the players from different teams are holistically different even if they perform the same actions. The players in the blue rectangle both perform the action of “standing”, but the holistic similarity between them is smaller than that between them and the player in the same team but with the different actions. Figure 1c,d are different activities of “queuing” and “talking”. However, the individual actions in both figures are standing, and they are holistically similar. The principal difference focuses on the interaction of an individual’s head and hand, as shown in the orange and green rectangles. Thus, utilizing more fine-grained part information and modeling its spatial-temporal relations is of great significance for group activity recognition tasks.

Motivated by the above observation, we propose a novel scheme to represent a part-based graph from multiple viewpoints for group activity recognition. As shown in Figure 2, both the static and dynamic representations are utilized. The former is obtained from a single image using a 2D CNN, and the latter is from multi-frame images and optical flow fields using a 3D CNN, respectively. For each representation, the same part-based graph representation network is designed, whose objective is to model the fine-grained relationships based on parts and pass messages between different parts of an individual and the same parts of different persons. They are represented as an intra-actor graph and inter-actor graph, respectively. In order to explore the interaction among different parts of an individual, we propose an intra-actor part graph module, which takes the part features as the node and automatically learns the weight of the edges. Additionally, the part features are refined by a graph convolution operation. To explore the interaction among the same parts of different individuals, we propose an inter-actor part graph module to capture part-level interactions for group activity. It is able to explore a more fine-grained relationship representation. The node in the inter-actor part graph is not a feature vector but a feature map which preserves the spatial pattern appearance of the individuals. Then, we use a visual mutual attention mechanism to construct the visual relation graph based on the appearance similarity and the location relation graph based on the relative distance between individuals, respectively. Finally, the information is passed between the part regions of different individuals by graph convolution.

The contributions of this paper are as follows:We propose the part-level relation modeling method to explore the fine-grained interactive representations for intra-actor and inter-actor parts.An intra-actor part graph module is designed to explore structural information of individuals for discriminative individual representations based on the interaction between different parts of a single individual, and an inter-actor part graph module is designed to explore the latent part-level context information by capturing the visual and location relation between the same part features of different actors.The experimental results on two publicly available datasets, the Volleyball and Collective Activity datasets, demonstrate that the proposed method achieves state-of-the-art performance.

In the following sections, we first introduce related works on group activity recognition and action recognition in Section 2 and then focus on the algorithm implementation and operation process of our work in Section 3. Next, in Section 4, we introduce the ablation study results, comparative experimental results, and visualization results in detail. In Section 5, we summarize the conclusions and future research.

## 2. Related Work

### 2.1. Group Activity Recognition

Group activity recognition is one of the most significant tasks in the field of the computer vision and multimedia community. Initially, earlier approaches tackled this task mostly based on hand-crafted features, which are then processed by probability graphical models to capture interaction relationships [4,5,6,20,21]. As deep learning approaches have become increasingly popular in recent years, group activity recognition has rapidly developed. Some proposed methods used spatial-temporal information to model the individual’s interactions and capture the temporal dependencies by RNN or LSTMs [14,15,22,23,24,25]. Ibrahim et al. [14] proposed a two-stage hierarchical deep model to model the temporal dynamics of individual actions and aggregate the individual-level information into a group-level feature. Yan et al. [15] used Bi-LSTM and Aggregation LSTM to model the long motions and flash motions of actors. Deng et al. [22] integrated a graphical model and deep neural network into a joint framework and used an RNN network to infer temporal information. Bagautdinov et al. [26] presented a unified framework for multi-people detection and group activity recognition. A matching mechanism is designed for associating the boxes in consecutive frames with the help of an RNN.

Recently, the rise of graph convolution networks (GCN) [27], attention mechanisms [28], and transformer networks [29] has further promoted the field of group activity recognition. Gavrilyuk et al. [16] extracted an individual’s static features and dynamic features by HRNet [13] and a 3D CNN and utilized the transformers to aggregate features by a self-attention mechanism. Wu et al. [17] built a relation graph and captured the appearance and position interaction relations among actors by graph convolution. Lu et al. [30] created a graph attention interaction model to explore potential interaction relationships between individuals and subgroups. Similar to [17], we also rely on graph representations, but differently, we explore the interactions among part-level features, which contain detailed information of individual representation.

### 2.2. CNN for Action Recognition

With the rapid progress of deep learning technology, the application of CNNs is becoming more and more extensive. For example, CNNs are used for image classification/segmentation [11,31], object/face detection [9,32], speech recognition [33], intelligent judge systems [34], smart cities safety improvement [35], and so on. However, traditional 2D CNNs have some limitations which cannot be applied in video streams. To solve this problem, some deep models have been proposed and applied in action recognition [36,37,38,39,40]. In 2014, Simonyan et al. [41] designed a two-stream convolutional network which can capture temporal and spatial dimensions dependently for video recognition. Carreira et al. [36] proposed a new network architecture named “Two-Stream Inflated 3D ConvNet (I3D)” in 2017. Currently, it is a widely used backbone for action recognition due to its expressive performance. In addition, convolutional neural networks not only can find the appearance features of images, but they can also learn the object’s motion relationship between adjacent video frames. Some existing works demonstrate the powerful ability of the optical flow for representing motion information in videos [42]. In this paper, we utilize a 2D CNN, a 3D CNN, and flow representations to combine actors’ static features and dynamic features for group activity recognition.

## 3. Methodology

### 3.1. Overview

The interaction between different parts can provide better context information for group activity recognition. The objective of the proposed scheme is to build a group activity recognition framework that can represent a more fine-grained relationship based on part-level features. The framework is shown in Figure 2.

The proposed scheme involves four steps: (1) part feature extraction for actors; (2) part interaction modeling at the intra-actor level; (3) part interaction modeling at the inter-actor level; and (4) individual action and group activity classification. Firstly, a 2D CNN and two 3D CNNs are used to extract both the static and dynamic features. Two 3D CNNs separately take the optical flow fields and RGB images as input to capture the motion information. The feature maps of each actor can then be directly extracted by RoIAlign [43]. Secondly, graph neural networks are constructed to model the intra-actor part relationships and inject the structural information of the human body into the part features. Thirdly, another kind of graph neural network is added to model the inter-actor part relationships. Finally, the embeddings involving the two-fold part-level interactions are applied for individual action and group activity classification. Different classification scores from static and dynamic branches are fused to obtain the final results.

### 3.2. Part Feature Extraction

Given video frames with bounding boxes indicating the locations of actors, we aim to extract the part features for each actor.

There are various features in video frames for group activity recognition. Inspired by the two-stream network [41], we employ two main branches to extract the static and dynamic features, respectively. In the static branch, a 2D CNN called Inception v3 [44] is used for a single frame. In the dynamic branch, a kind of 3D CNN called I3D [36], which is proved to have excellent performance after pre-training, is used for sequential frames. Since the motion information between frames is more difficult to capture, we exploit two different I3D networks that separately act on the RGB images and the optical flow fields pre-computed from consecutive frames. As a result, there are three backbones in the proposed framework.

With backbones truncated at certain intermediate layers, we can obtain the feature maps containing both appearance features and location information for all actors. Taking advantage of the location information and input bounding boxes, we perform RoIAlign to obtain the feature maps for each actor. Assume the individual feature maps have a size of k×k and channel of *D*. Different from previous methods that mostly transform the feature maps into the feature vectors for recognition, we treat the individual feature maps as a set of part features $f_{n}={\left\{f_{n}^{p}\right\}}_{p=1}^{p=k\times k}$, where $f_{n}^{p}\in R^{D}$ denotes the *p*-th part feature with dimension of *D* for the *n*-th actor. In this way, no extra computation is required, and the parts can provide more detailed appearance information for the fine-grained individual action and group activity classification.

### 3.3. Intra-Actor Part Graph Module

In existing methods [17,45], the performance of individual action recognition is still low, which further hinders the improvement of group activity performance. These methods have ignored the structural information of individuals, which is helpful for individual action representation. The relationship between different parts of the individual helps to understand the individual action. Therefore, we establish an intra-actor part graph module for each individual with shared parameters to explore the structural information. To explore the relations of parts, we use GCN [27] to model the relations between parts. We first revisit the traditional GCN and then present our intra-actor part graph module in detail.

Revisiting Graph Convolution Network. A graph convolution network (GCN) is a kind of convolutional neural network that can operate directly on irregular graph data and has the ability to encode graph-structural information. It is suitable for many computer vision tasks, such as skeleton-based action recognition, traffic forecasting, and person re-identification. GCN operation can be expressed as follows. Firstly, input a set of features as nodes $X\in R^{N\times D}$; *N* is the number of nodes, and *D* represents the node feature dimensions. Secondly, construct adjacent matrix $A\in R^{N\times N}$ to represent pair-wise relation among nodes. Finally, *A* and learnable weight matrix *W* are utilized to aggregate neighborhood information by message passing mechanism, and obtain enhanced feature representation *Y*. The graph convolution formula can be represented as follows:(1)Y=δ(AXW)
where *X* represents node features and *W* is learnable weight matrix, *A* represents a graph structure that is pre-defined or constructed by dot-product operation between two node features. The value of *A* represents the strength of correlation. δ is the activation function.

Intra-Actor Part Graph Module. The part graph module automatically learns the weights of the edge between each node pair and passes the message across intra-actor parts to refine part features. Finally, the part graph module outputs the refined feature map for the subsequent inter-actor interactions.

As shown in Figure 3, to model the relationship, a fully connected graph $G_{part}=\left\{V_{part},E_{part}\right\}$ is firstly established, where $V_{part}$ is the node set of the graph, corresponding to all the part features of the *n*-th actor $f_{n}={\left\{f_{n}^{p}\right\}}_{p=1}^{p=k\times k}$; $E_{part}$ is the set of edges. To represent the importance of different parts, we calculated the weight of the edge between each pair of parts as follows:(2)$$e_{n}^{ij}=h\left[\phi f_{n}^{i};\phi f_{n}^{j}\right]$$
where $e_{n}^{ij}$ denotes the edge weight between the *i*-th and *j*-th parts for the *n*-th person. It represents the correlation between two parts; $\phi \in R^{D^{\prime }\times D}$ is a symmetrical transformation; $D^{\prime }$ is its embedded dimension; [;] represents the concatenation operation between features; and $h\in R^{1\times 2D^{\prime }}$ is a weight matrix.

In this way, for the *i*-th part of the *n*-th person, the weights between every neighbor $j\in N_{i}$ are computed, where $N_{i}$ is the set of neighbors of part *i*. Finally, we normalized the weights across all neighbors of part *i* as follows:(3)$$A_{n}^{ij}=\frac{exp\left(\sigma \left(e_{n}^{ij}\right)\right)}{\sum_{u\in N_{i}}exp\left(\sigma \left(e_{n}^{iu}\right)\right)}$$
where σ is the LeakyReLU activation function [46], and $A_{n}^{ij}$ is the normalized weight between the part *i* and *j* of the *n*-th individual.

Once the graphs are built, we can perform relational reasoning among intra-actor parts by a single-layer GCN. The refined features contextualize with structural information of the individual. In addition, we find that introducing a residual connection is helpful for training the a stable network. The calculation of graph convolution for the *i*-th part feature of the *n*-th individual can be represented as
(4)$$f_{n}^{{}^{\prime }i}=ReLU\left(A_{n}^{i}f_{n}^{i}W\right)+f_{n}^{i}$$
where *W* is a shared linear transformation function, and $f_{n}^{{}^{\prime }i}$ represents the refined part features, all of which are integrated into a refined feature map that has the same shape as the original feature map and preserves the spatial pattern. The refined feature map is the input for the following inter-actor part graph module.

### 3.4. Inter-Actor Part Graph Module

The objective of the inter-actor part graph module is to mine the interaction relationships of parts among individuals. The existing method used the global visual features of individuals’ output from the fully connected layer to model the relationship, but the global features lost the local spatial information of individuals. In the antagonistic sports video, in particular, the same team players wear similar suits, the holistic visual characteristics are similar, and their actions mainly depend on the more detailed information of posture. The loss of detailed information will result in performance degradation for individual actions and group activity. Therefore, we propose an inter-actor part graph module and conduct a more fine-grained relationship modeling at the part level.

As shown in Figure 4a, in the conventional graph convolution network [27], the nodes represent the vectors, and the edge is a scalar value to represent the weight between the nodes. In our proposed inter-actor part graph, the nodes are the feature maps of individuals, and the edge is a vector in which each element represents the interactive weight of each part. Specifically, the node set in the inter-actor part graph corresponds to the feature maps of actors $F=\{f_{i}|i=1,\ldots ,N\}$, where *N* is the number of actors in the scene. We constructed graphs $G\in R^{N\times N\times P}$ to represent the part features relationship among individuals, where *P* denotes the number of parts of an actor, and the relation vector $G_{ij}\in R^{P}$ denotes the importance of each part of actor *j* to actor *i*, as in Figure 4.

To increase the ability of relationship representation between individuals, two kinds of relations are considered: visual relation and location relation. As the visual relation and location relation belong to different semantic characteristics, we model these two relations separately. Our inter-actor part graph module is composed of two kinds of relation graphs, which will be introduced in detail next.

Visual relation graph. Each individual is represented as a feature map with a size of k×k in which each cell represents a part feature. When modeling the interaction of inter-actor parts, we need to explore the mutually important and relevant features among parts of different individuals. To solve this problem, we learn from the mutual attention mechanism [47,48], which is originally introduced in social networks. The mutual attention mechanism is also used to calculate the edge weight vector in the visual graph in our work.

To model the visual relationship between *i*-th actor and *j*-th actor, we first pair their feature maps $f_{i}$, $f_{j}\in R^{D\times P}$, where *P* is the number of parts for an actor; *D* is the dimension of each part feature. Then, a mutual attention matrix $\;A\in R^{D\times D}$ is introduced to calculate the part correlation matrix $M^{ij}\in R^{P\times P}$:(5)$$M^{ij}=tanh\left({\left(f^{i}\right)}^{T}Af^{j}\right)$$
where the mutual attention matrix *A* is a shared matrix among individuals. The part correlation matrix $M^{ij}$ represents the correlation between each part of the *i*-th actor and the *j*-th actor. Each element $M_{a,b}^{ij}$ represents the pairwise correlation score between two part features $f_{a}^{i},f_{b}^{j}\in R^{D}$ from actor *i* and actor *j*. Similar to [47], we next perform average pooling on each column of $M^{ij}$ to generate part importance vector $S^{ij}\in R^{P}$, whose elements represent the weight of each part in actor *j* to actor *i*:(6)$$S_{p}^{ij}=\frac{1}{P}\sum_{k=1}^{P}M_{k,p}^{ij}$$

Then, the softmax function is used to normalize the part importance vector to generate the final relation vector across the same parts in different individuals so that the sum of the weights of all individuals at the same part is 1. We calculated the *p*-th element in relation vector $G^{ij}$ as follows:(7)$$G_{p}^{ij}=\frac{exp\left(S_{p}^{ij}\right)}{\sum_{k=1}^{N}exp\left(S_{p}^{ik}\right)}$$

Location relation graph. In some group activities, such as sport activities, talking in surveillance videos, and so on, the relationship between individuals is highly correlated with their relative distance. The closer the two individuals are, the stronger the relationship between them may be. On the contrary, the farther the distance is, the weaker the relationship will be. Therefore, we use the relative distance between individuals for building the graph.

Similar to [49], given the position coordinates of *N* individuals in the video $B={\left\{b_{n}\right\}}_{n=1}^{N}$, where $b_{n}=\{x_{n,mid}$,$y_{n,mid}\}$. $x_{n,mid}$ and $y_{n,mid}$ represent the center coordinates of the *n*-th actor along *X*-axis and *Y*-axis, and they are normalized as follows:(8)$$c_{n}=\left(\beta \frac{x_{n,mid}}{W},\beta \frac{y_{n,mid}}{H}\right)$$
where $c_{n}\;$ is the normalized center position of the *n*-th actor; β is a scale coefficient. We set it to 10 empirically. *W* and *H* are the width and height of the video frame, respectively. Then, we defined the location correlation score $a_{ij}$ based on the relative distance as
(9)$$a_{ij}=exp\left(-\frac{{\|c_{i}-c_{j}\|}_{2}^{2}}{2}\right)$$

In this solution, if the *i*-th and *j*-th individuals are close to each other in space, the corresponding $a_{ij}$ will be larger, and vice versa.

In order to extend the location correlation score to the part level, a simple way is used to set the location relation score for each part of the *j*-th actor to the *i*-th individual as $a_{ij}$:(10)$$S_{p}^{ij}=a_{ij},\;p=1,\ldots ,P$$

Finally, the same normalization operation used in the visual relation graph is conducted to $S_{p}^{ij}$, and the part relation vector $G^{ij}$ based on the relative distance can be obtained. In this way, the spatial information is embedded into the part relation vector.

Reasoning of Inter-Actor Part Graph. After establishing the inter-actor part graph, it is necessary to conduct relationship reasoning on the graph to obtain relation features for individual action and group activity recognition. Motivated by the graph convolution network (GCN) [27], which aggregates node features from neighbors according to the weights of edges, we defined the rule of message passing in the inter-actor part graph as follows:(11)$$X_{p}^{i}=\sigma \left(\sum_{j=1}^{N}G_{p}^{ij}X_{p}^{j}W\right)$$
where $X_{p}^{i}$ is the *p*-th part relation feature of actor *i* after interactive updating with other individuals, $G_{p}^{ij}$ is the part weight of *p*-th part as aforementioned, and $X_{p}^{j}$ is the *p*-th part of other actor *j* in the neighborhood of the *i*-th actor. *W* is a shared weight matrix, and σ(·) denotes an activation function. The aggregation computation is performed between the same parts of the different actors because the same part may contain similar semantic information, such as the heads, hands, and legs of different people. The correlations between them are stronger, and the message passing is more effective.

The original GCN network contains a single kind of graph. In contrast, our inter-actor part graph includes two types of graph structures: visual relation graph and location relation graph. According to our experiment, we fuse outputs of two graphs by element-wise sum. Inspired by the residual network, the relational part features are combined with refined part features from the intra-actor part graph by summation to form the final actor part representation.

### 3.5. Fusion of Multiple Branches

Previous works have proved that the fusion of multiple branches holding complementary information can effectively improve the performance for action recognition [16,50]. Inspired by this, we integrate multiple representations into a framework. We explore the impact of the branches with dynamic temporal information and static spatial information on performance. The static spatial branch uses Inception-v3 [44] to extract the individual features from a single frame, and the dynamic temporal branch uses I3D network [51] to extract the spatial-temporal features of actors. Because RGB and optical flow can represent different aspects of dynamic information, two different kinds of inputs are considered. To fuse the static and dynamic branches, similar to the literature [41], we use the late fusion strategy to sum the predicted scores of each branch, and we set the weight for the dynamic branch to be twice as large as the static branch, following [41]. The advantage of late fusion is that each branch is independent of others and can concentrate on modeling by static and dynamic representations.

### 3.6. Training Objective

We embed the feature map of each actor output by the inter-actor part graph to a *d*-dimensional vector representation by a fully connected layer, which is used to classify individual action. All *N* actors in the scene are maxpooled to obtain the *d*-dimensional group-level feature, which is used for group activity classification. The whole model can be trained in an end-to-end way using cross-entropy loss as follows:(12)$$L=\lambda_{1}L_{1}\left(y_{g},{\hat{y}}_{g}\right)+\;\lambda_{2}L_{2}\left(y_{i},\hat{y_{i}}\right)$$
where $L_{1}$ and $L_{2}$ are the cross-entropy loss functions; $y_{g}$ and $y_{i}$ are the groundtruth labels; and ${\hat{y}}_{g}$ and ${\hat{y}}_{i}$ are the prediction results of group activity and individual action classification. $\lambda_{1}$ and $\lambda_{2}$ are weights to balance the two tasks.

## 4. Experiments

In this section, we present experiments on two widely used benchmarks: the Volleyball Dataset [14] and Collective Activity Dataset [52]. We first introduce the datasets and the implementation details of our method. Then, we conduct ablation studies to validate the effects of the proposed modules in our method. Lastly, we compare the performances of our model with the state-of-the-art methods and provide a visualization analysis of the results.

### 4.1. Datasets

The Volleyball Dataset (VD) [14] contains 4830 clips (3493 training clips and 1337 testing clips) from 55 videos of a volleyball game. Every clip has eight group activity labels (including spiking, setting, passing, and winpoint in the left or right court), and 9 individual action labels (including waiting, setting, digging, failing, spiking, standing, jumping, moving, and blocking). The individual action categories differ from group activity categories.

The Collective Activity Dataset (CAD) [52] consists of 44 video sequences. The length of each sequence ranges from 190 frames to 1800 frames. We follow the experiment setting of [2] and use 32 videos for training and 12 videos for testing. The dataset contains five group activities (crossing, walking, queuing, talking, and waiting). The individual action labels have six categories, including NA, crossing, walking, queuing, talking, and waiting. The individual action categories are basically consistent with group behavior categories.

### 4.2. Implementation Details

Following related works, we use the middle frame, five frames before it, and four frames after it as the input to our model on both datasets. Inception-v3 and I3D are used as the backbone for the static network and dynamic network, respectively. For the static network, following [16], we randomly sample a single frame from each clip during training and use the middle frame of the input video during testing. The size of the feature map extracted by RoIAlign, *k* is set to 3 for the static and dynamic branches. In addition to the RGB stream of the dynamic network, we also use optical flow fields computed by the TVL1 algorithm [53] as input to another stream. The dimension of the feature vector for classification *d* is set to 1024. For the training loss, we set $\lambda_{1}=\lambda_{2}=1$. We use the Adam optimizer [54] to train the network with fixed hyperparameters to $\beta_{1}=0.9$, $\beta_{2}=0.999$, $\epsilon =10^{-8}$. Moreover, we use two metrics for evaluating our framework performance: MCA (%), which represents multi-class classification accuracy; MPCA (%), which represents mean per class accuracy; and a confusion matrix. With the help of RoIAlign, matplotlib, numpy, and torchvision machine learning-based libraries, all experiments are performed based on the Pytorch deep learning framework, Pycharm IDE, and a single GeForce RTX 2080 GPU.

Volleyball Dataset. For a fair comparison, we use the tracklets generated by [26]. Input images from the VD are resized to 720×1280. We follow [14] to divide players into two groups according to their position to eliminate the confusion of the left team and right team activity. For the training of the static network, we train the network in 250 epochs using a minibatch size of 4 and a learning rate ranging from 0.0003 to 0.00001. For the training of the dynamic network, we train the network in 70 epochs and use a minibatch size of 2 with a learning rate ranging from 0.0002 to 0.00001.

Collective Activity Dataset. Images from CAD are resized to 480×720. For the training of the static branch, we use a minibatch size of 4 and a learning rate ranging from 0.00002 to 0.000001 to train the network. For the training of the dynamic branch, we use a minibatch size of 8 and trained the network in 70 epochs with a learning rate set to 0.0002.

### 4.3. Metrics

MCA/MPCA. MCA represents the percentage of correct predictions. MPCA represents the average accuracy for each class.
(13)$$\mathrm{MCA}=\frac{M_{c}}{M}$$
where $M_{c}$ is the total number of correctly classified samples, and *M* is the total number of samples.
(14)$$\mathrm{MPCA}=\frac{1}{N_{G}}\sum_{i=1}^{N_{G}}\frac{M_{c}^{i}}{M^{i}}$$
where $N_{G}$ is the number of categories, $M_{c}^{i}$ represents the total number of correctly classified samples in category *i*, and $M^{i}$ is the total number of samples in category *i*.

Confusion matrix. The confusion matrix reflects the classification accuracy of each category, which can display the classification effect of each category. Each column represents the ground-truth labels, while each row represents the predicted classes, and the values reflect classification accuracy.

### 4.4. Ablation Study

To prove the effectiveness of each component of our proposed model, we perform an ablation study on the Volleyball Dataset and Collective Activity Dataset using multi-class classification accuracy (MCA) as the evaluation metric.

Effectiveness of part-based graphs network. To validate the influence of the part-based graphs network, a variant is implemented by removing the inter-actor part graph and the intra-actor part graph modules as the simplest baseline, which is denoted as the “Non-Part Graph”. More specifically, we use Inception-v3 pretrained on ImageNet as the backbone for the static network and I3D pretrained on Kinetic-400 for dynamic networks. The networks are fine-tuned on the Volleyball Dataset using cross-entropy loss for individual action and group activity, respectively. As shown in Table 1, the static branch results of MCA on the Volleyball Dataset and Collective Activity Dataset reach 92.3% and 88.9%. In comparison with the results of “Non-Part Graph” models, the MCA scores are improved by a margin of 1.5% and 2.9% on the two datasets. The dynamic branch results of MCA are improved by about 2% and 3.7%. Thus, our final models of static and dynamic branches perform better than non-part graph models on the two datasets, which demonstrates the advantages of our part-based graphs network. These experimental results prove that the part feature modeling is more effective than individual-level features in acquiring spatial-temporal relationship information. In our paper, the interaction between parts is divided into two types: the inter-actor and the intra-actor. In particular, the intra-actor part graph models the structural information of individuals, and the inter-actor part graph explores the relational context information among individuals for reasoning group activity. In contrast, the “Non-Part Graph” model neglects rich interactive information, resulting in an obvious decline in performance. These results also reflect that it is necessary to use fine-grained structural information and construct their spatial-temporal relations for group activity recognition.

Effectiveness of Intra-Actor Part Graph and Inter-Actor Part Graph. To evaluate the contributions of these proposed modules, we design four variants. w/o Intra-Actor Part Graph: this variant removes the intra-actor part graph and constructs the inter-actor part graph by original appearance features. w/o Inter-Actor Part Graph (V+L): this variant unloads the visual graph (V) and location graph (L). The features refine by the intra-actor part graph are directly used for classification without exploring interactions among actors. w/o Inter-Actor Part Graph (V): we only unload the visual graph in this variant, which is implemented to understand the effect of the visual graph for group activity inference. To examine the efficacy of the location graph, a variant without this component is studied, which is denoted by w/o Inter-Actor Part Graph (L).

As shown in Table 1, our final model consistently outperforms these variants of static and dynamic branches on both datasets by learning part interaction in intra-actor and inter-actor simultaneously. This indicates that exploring relationships among parts is beneficial for group activity recognition. By comparing the results of “w/o Inter-Actor Part Graph (V+L)” and “w/o Intra-Actor Part Graph” in Table 1, the MCA(VD)-S and MCA(VD)-D are improved by about 0.5% and 0.7% on the Volleyball Dataset. As for the Collective Activity Dataset, the results of the static branch increased from 86.9% to 87.9%, and the dynamic results of “w/o Intra-Actor Part Graph” reaches 89.6%, improved by about 1% compared with “w/o Inter-Actor Part Graph (V+L)”. These results indicate that “w/o Intra-Actor Part Graph” slightly outperforms “w/o Inter-Actor Part Graph (V+L)”, which also suggests that the inter-actor part graph is able to bring more information cues for inferring group activity.

We also observe that the MCA of “w/o Inter-Actor Part Graph (V)” reaches 91.4% in the static branch and 92.7% in the dynamic branch on the Volleyball Dataset, which indicates that the visual relation graph will affect about 0.9% and 0.6%, respectively. The MCA of “w/o Inter-Actor Part Graph (L)” on the Volleyball Dataset is lower than our final model by about 0.9% and 0.9% in two branches. These results indicate that the location graph is more important than the visual graph in the dynamic branch. As for the Collective Activity Dataset, the conclusions are similar to those obtained from the Volleyball Dataset. The MCA of “w/o Inter-Actor Part Graph (V)” or “w/o Inter-Actor Part Graph (L)” perform lower than our final model, which indicates that the visual graph and location graph can complement each other.

The impact of number of parts in part graphs. In this experiment, we evaluated the results in the dynamic network for RGB as input when *k* = 2, 3, 4, 5. *k* is the crop size of the feature map extracted by the RoIAlign layer, and the number of parts in the part graph is k×k. As shown in Table 2, the model has the best performance when k=3, which can reach about 93.3% on the Volleyball Dataset and 91.6% on the Collective Activity Dataset. When the number of parts is too small or large, the performance will be reduced. The reason is that the image will be divided too delicately and can not contain effective information when it is too large. On the contrary, when the number of parts is too small, subtle but discriminative information may be ignored.

### 4.5. Comparison to the State of the Art

Volleyball Dataset. We compare our method with the state-of-the-art approaches on the Volleyball Dataset in Table 3. G-MCA and I-MCA respectively denote the accuracy of group activity recognition and individual action recognition. For a fair comparison, we report the result of ACRF without using AlphaPose [55] obtained in [45]. We report two variants of our model, the fusion of two dynamic branches of RGB and flow input and the fusion of two dynamic branches with the static branch. The proposed method has a competitive result with the best-performing methods of [16,45,56] and surpasses other approaches [15,17,50,56,57,58,59,60]. On the Volleyball Dataset, our method can achieve better results, mainly because the spatial-temporal relation information based on part features is more discriminative than modeling based on individual features. Our method surpasses ACRF [45], which achieved the best performance on the Volleyball Dataset, by 0.3% and 1.9% for group activity and individual action recognition, respectively. With the same I3D backbone using the late fusion of RGB and flow representation, our method improves the performance of the actor-transformer [16] more than 1.1% and 0.4% with respect to the accuracy of group activity and individual action and outperforms the CRM [50] by 1.1% for group activity recognition.

Collective Activity Dataset. We compare the proposed method with previous methods on the Collective Activity Dataset. Firstly, we compare the results of the multi-class classification accuracy (MCA) of our model. The results of each branch are as follows: Inception v3—88.9%, I3D RGB—91.6%, and I3D Flow—82.3%. It is worth mentioning that our single dynamic branch of RGB input already outperforms many existing methods [2,17,57]. We explore the late fusion of dynamic and static branches, reaching a 93.2% group activity recognition accuracy, slightly outperforming most existing state-of-the-art methods, which demonstrates its effectiveness. However, ACRF achieves a better result than our model since they adopt ResNet50, a stronger backbone network, and utilize FPN [62] to improve the ability of feature representation. Several works [23,50] have argued that the walking class is ill-defined. Therefore, we merge the “walking” and “crossing” classes as “moving” and report the mean per class accuracy (MPCA) to evaluate the performance, following [23]. The results of some methods [2,16] are calculated from the confusion matrix reported in their paper. According to the results, our model outperforms other approaches in this setting. The results indicate that the most incorrect predictions of our model are between walking and crossing activities.

### 4.6. Visualization

To further understand our model, we visualized the relations captured by the visual relation graph in Figure 5. Numbers of 0 to 11 are assigned to 12 actors respectively in Figure 5a. We summed part-level relation values of actors in the relation vector to a scalar, indicating the global relation weight among actors. We describe the pairwise global relation weights as a N×N adjacency matrix, which is shown in Figure 5b. We can see that our model can mine high relations between actors 5 and 8, who are blocking and spiking, as is shown in columns 5 and 8 of the matrix. Our model can discover the relevant actors who determine the group activity. Furthermore, our model is able to suppress the actors who are standing and have almost no relation to the others, as shown from columns 0 to 4 in the matrix, because these actors in the left team do not participate in the right spike activity. We also illustrate two examples of the weight vector learned by the visual relation graph in Figure 5c, and it shows that our model can capture effective relation information for group activity recognition. Each element of the weight vector indicates the importance of the corresponding part to the others. Actor 5 who is in the blocking action has higher relation values in the upper-body part than actor 8 since his arms can provide more visual cues. Actor 11, who is irrelevant for group activity, has small values for all the parts.

Additionally, we draw the confusion matrix based on our model to analyze the performance, as shown in Figure 6 and Figure 7. As seen from these figures, our method can achieve a recognition accuracy over 90% in terms of all of the group activities on the Volleyball Dataset. Nevertheless, it shows confusion between set and pass as two kinds of activities having similar interaction relationships. Compared with the confusion matrix results of HIGCIN [1], stagNet [2], and HDTM [14], our method has the best accuracy in each category. The main reason is that we use the representations of parts that are more correlated with group activity to conduct spatial-temporal modeling; this method makes it more effective to find associations between specific parts of the body and group activity. For the Collective Dataset, our model is easily confused by “walking” and “crossing”: about 17% of the samples in “crossing” are misclassified as “walking”. The reason for the false classification is that these two kinds of activities have a similar appearance and spatial-temporal relationship information. Moreover, “walking”, “crossing”, and “waiting” always appear together in the same scene, such as a traffic intersection, which will affect the performance of our model.

## 5. Conclusions

In this work, we propose a multi-perspective representation to a part-based graph method for group activity recognition to explore the relationship between intra-actor and inter-actor parts. More specifically, we propose an intra-actor part graph module to learn structural information of individuals and an inter-actor part graph to capture the visual relation and location relation simultaneously, exploring latent part-level context information. The compared evaluation on two standard benchmarks demonstrates that the proposed method achieves competitive results compared to state-of-the-art methods. These results indicate that our module can effectively and accurately complete the relationship modeling in body part-level features. Moreover, the fine-grained structural information relationship modeling plays an important role in improving classification tasks. This research can be further extended to pedestrian search, vehicle/pedestrian re-identification, and other scenarios. In the future, we will explore how to combine fine-grained information and semantic information effectively.

## Figures and Tables

**Figure 1 sensors-22-05521-f001:**
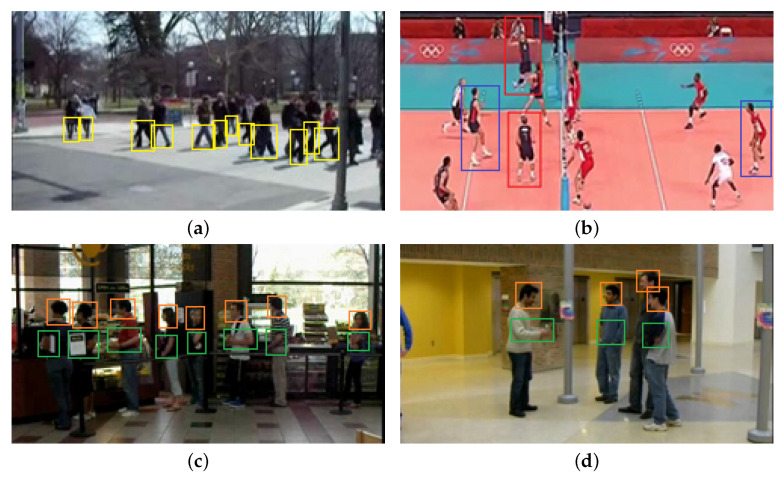
Illustration of the importance of the local parts. (**a**) “walking” is significantly related to the legs. (**b**) The players in the same team have the similar holistic appearance although they perform different actions, while thoses in the different teams have the holistically different but locally similar appearance when they perform the similar actions. The group activities in (**c**,**d**) are queuing and talking, respectively. The persons in the two pictures look similar in terms of most body parts. The interactions of their heads and hands are critical cues.

**Figure 2 sensors-22-05521-f002:**
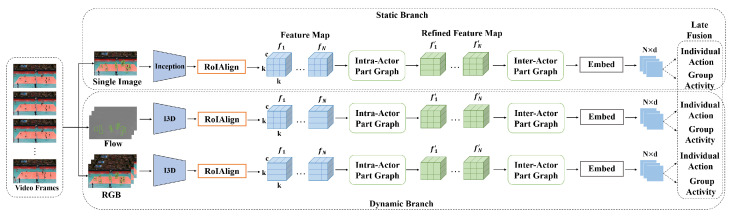
An overview of the proposed framework for group activity recognition. It consists of two branches for modeling the static and dynamic representations. For the two branches, we apply RoIAlign to extract feature maps of actors, and we decompose them as a set of parts. Both branches incorporate an intra-actor part graph module and inter-actor part graph module to explore intra-actor and inter-actor interactions in the part-level features. Finally, we embed the part features into feature vectors and feed them into classifiers for individual action and group activity recognition. The prediction scores of different branches are late fused to obtain the final results.

**Figure 3 sensors-22-05521-f003:**
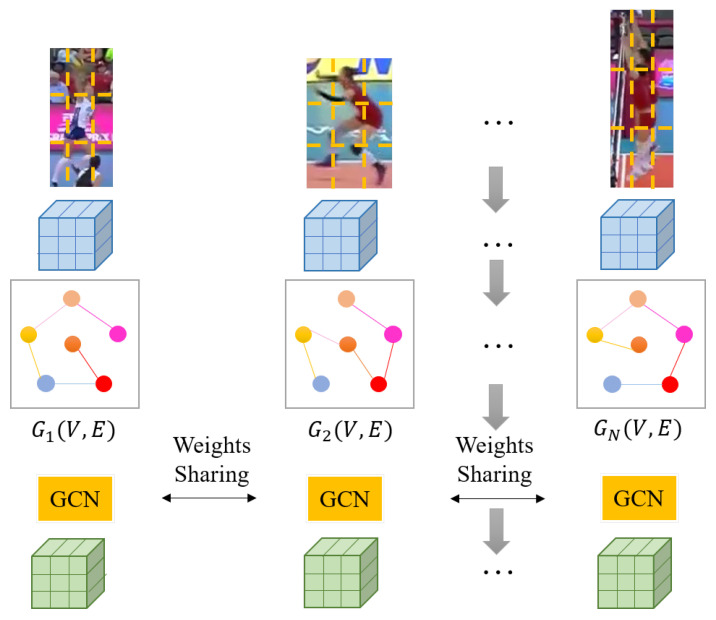
Illustration of the intra-actor part graph module. The blue and green cubes mean the original feature map extracted by backbone and the refined feature maps, respectively. The feature maps can be treated as a set of part features. We build graphs for each individual, and the nodes in different colors represent different parts. GCN is applied for message passing among parts to refine original feature maps. To reduce the parameters, the weights in intra-actor part graph module are sharing for all individuals.

**Figure 4 sensors-22-05521-f004:**
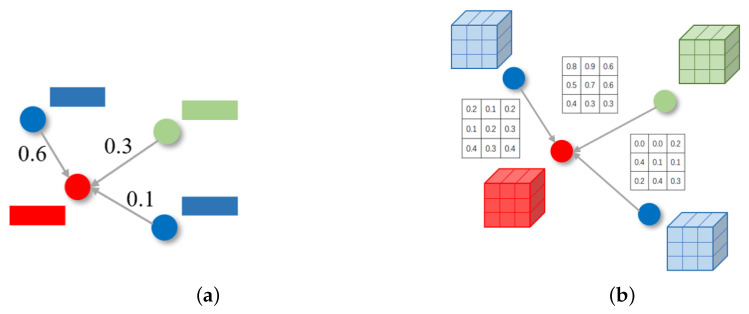
The difference of graph construction between (**a**) conventional graph network and (**b**) our proposed inter-actor part graph. The different colors represent different individuals in the scene.

**Figure 5 sensors-22-05521-f005:**
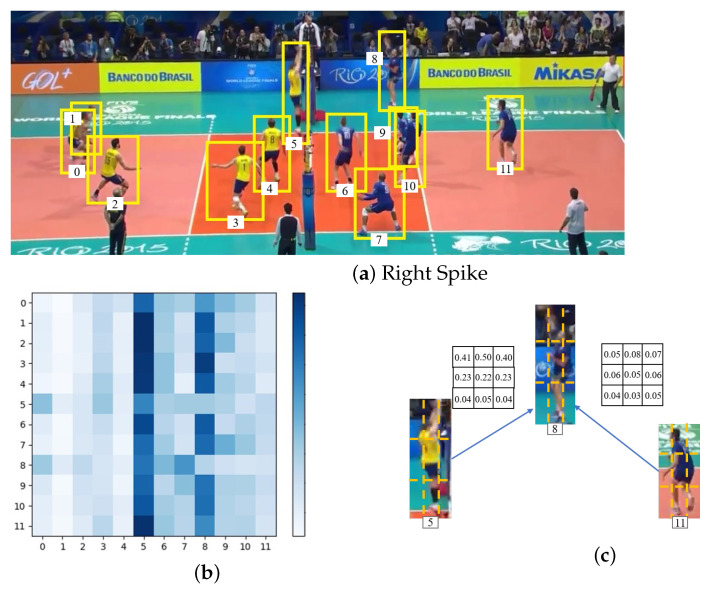
Visualization of part-level inter-actor interaction relations captured by the visual relation graph on Volleyball Dataset. (**a**) Input frame with ground-truth bounding box. Each actor is assigned one of numbers in [0, 11]; (**b**) N×N dependency matrix. *N* denotes the number of actor in the video. The values change from large to small along with the colors changing from blue to white; (**c**) Examples of part-level weight vector.

**Figure 6 sensors-22-05521-f006:**
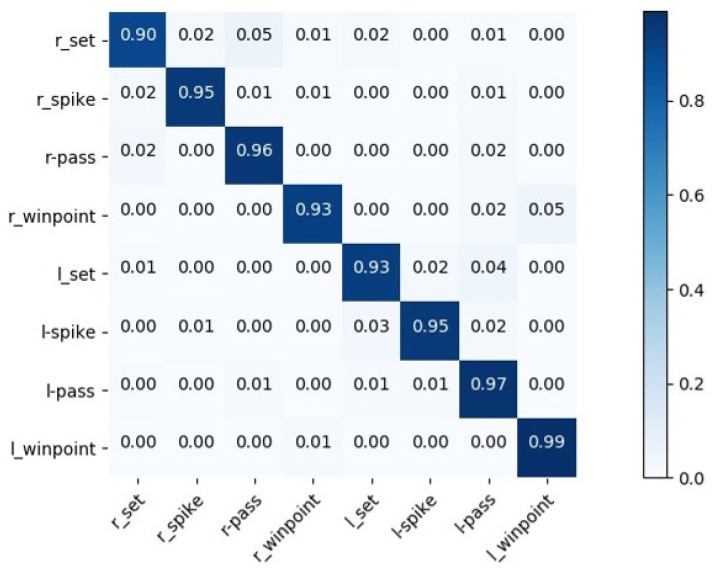
Confusion matrices for I3D + Inception-v3 model on Volleyball Dataset. “l-” and “r-” are abbreviations for “Left” and “Right” in the group activity labels.

**Figure 7 sensors-22-05521-f007:**
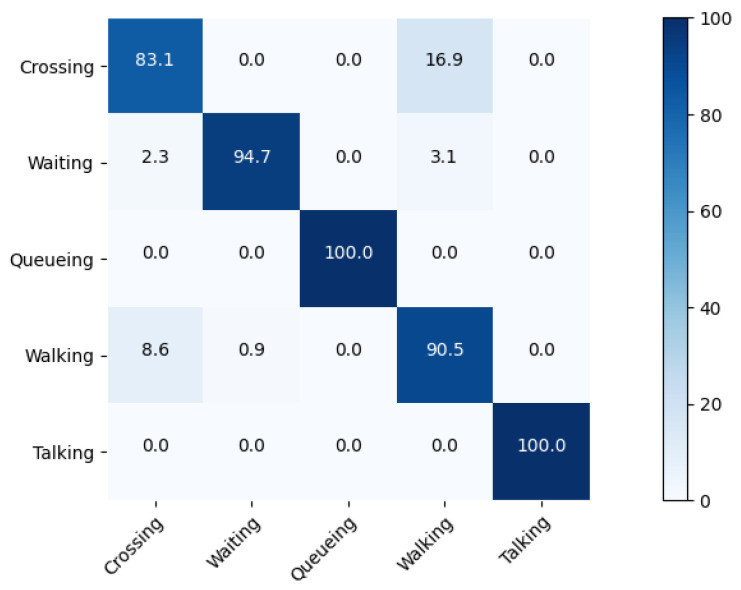
Confusion matrices for I3D + Inception-v3 model on Collective Activity Dataset.

**Table 1 sensors-22-05521-t001:** Performance comparison with different variants for the proposed method on Volleyball and Collective Activity datasets. ’-S’ denotes the results of the static branch; ’-D’ denotes the results of the dynamic branch.

Models	MCA(VD)-S	MCA(VD)-D	MCA(CAD)-S	MCA(CAD)-D
Non-Part Graph	90.8	91.3	86.0	87.9
w/o Inter-Actor Part Graph (V+L)	91.2	92.1	86.9	88.5
w/o Inter-Actor Part Graph (V)	91.4	92.7	87.1	90.7
w/o Inter-Actor Part Graph (L)	91.4	92.4	87.3	89.0
w/o Intra-Actor Part Graph	91.7	92.8	87.9	89.6
Ours(RGB)	92.3	93.3	88.9	91.6

**Table 2 sensors-22-05521-t002:** Performance comparison with different numbers of parts in our model on the VD and CAD.

Number of Parts	MCA(VD)	MCA(CAD)
4	92.7	89.3
9	93.3	91.6
16	92.9	91.0
25	93.2	90.3

**Table 3 sensors-22-05521-t003:** Comparison with the state-of-the-art methods on the Volleyball Dataset (VD) and Collective Activity Dataset (CAD). ‘G-’ and ‘I-’ denote `Group Activity Classification’ and `Individual Action Classification’.

Method	Backbone	G-MCA(VD)	I-MCA(VD)	G-MCA(CAD)	G-MPCA(CAD)
HDTM(RGB) [14]	AlexNet	81.9	-	81.5	-
PCTDM(RGB+Flow) [15]	AlexNet	87.7	-	-	92.2
CERN(RGB) [57]	VGG16	83.3	-	87.2	88.3
stagNet(RGB) [2]	VGG16	89.3	-	89.1	91.0
PRL(RGB) [58]	VGG16	91.4	-	-	93.8
HiGCIN(RGB) [1]	ResNet18	91.4	-	93.4	93.0
SPTS(RGB+Flow) [61]	VGG	90.7	-	-	95.7
SSU(RGB) [26]	Inception-v3	90.6	81.8	-	-
ARG(RGB) [17]	Inception-v3	92.5	83.0	91.0	-
CRM(RGB+Flow) [50]	I3D	93.0	-	85.8	94.2
Actor-Transformer(RGB+Flow) [16]	I3D	93.0	83.7	92.8	98.5
Actor-Transformer(Pose +Flow) [16]	HRNet + I3D	94.4	85.9	91.2	-
ACRF(RGB+Flow) [45]	I3D+FPN+ResNet50	94.5	83.1	94.6	-
STDIN(RGB) [59]	ResNet18	93.3	-	-	95.3
STBiP(RGB+Pose) [56]	HRNet+Vgg16	94.7	-	-	96.4
GroupFormer(RGB+Flow) [19]	I3D	94.9	84.0	94.7	-
Detector-Free(RGB) [60]	ResNet18	90.5	-	-	-
Ours(RGB+Flow)	I3D	94.1	84.1	92.2	98.8
Ours(RGB+Flow)	I3D+Inception-v3	94.8	85.0	93.2	98.8

## Data Availability

The Volleyball Dataset [14] is available on the website https://github.com/mostafa-saad/deep-activity-rec (accessed on 21 March 2022), and the Collective Activity Dataset [52] is available on the website https://cvgl.stanford.edu/projects/collective/collectiveActivity.html (accessed on 21 March 2022).

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
