# Peer review of "Multi-Perspective Representation to Part-Based Graph for Group Activity Recognition"

_sensors, 2022, doi:10.3390/s22155521_

Round 1

Reviewer 1 Report

Section 1 must be improved. You should introduce the problem in more detail so that the reader is immediately clear about the purpose of your study. You should add more information in the introductory part, you should add other works that have also addressed the problem. Your contribution is clearly stated, however before describing how you intend to deal with the problem you should adequately describe how other researchers have dealt with similar problems. There is a complete lack of an adequate bibliography that introduces the reader to the problem. Furthermore, there is a lack of adequate introduction to CNN. At the end of the section, add an outline of the rest of the paper, in this way the reader will be introduced to the content of the following sections.

Section 2 can be improved. I noticed successively that you inserted the introduction to the works of other researchers in the second section. I think it is appropriate to move this section to the Introduction. The introductory part of CNN should also be moved to the Introduction. Add more references to works that have already dealt with the topic (CNN), even if in different sectors, for example:” Improving smart cities safety using sound events detection based on deep neural network algorithms”.

Section 3 can be improved. This section is well written, I would try to work on the title trying to incorporate the term methodology. Also missing is a section on introducing the metrics you will use to evaluate the performance of your methodology.

Section 4 must be improved. In this section you present the results of your job. Unfortunately, the metrics used to evaluate the performance of your methodology have not been adequately introduced. In the tables it is not specified what the reader is reading, and the figures showing the matrices of confusions have not been preceded by an adequate description of the metric. Also you should describe in detail the software platform you used. Also describe the machine learning-based libraries you used. A detailed discussion of the results obtained is missing. Try to summarize what was obtained and try to extract useful information from the work carried out. Also add bibliographic references to support your conclusions, to give more weight to your statements.

Section 5 must be improved. Paragraphs are missing where the possible practical applications of the results of this study are reported. What these results can serve the people, it is necessary to insert possible uses of this study that justify their publication. They also lack the possible future goals of this work. Do the authors plan to continue their research on this topic?

117) Figure 2 must be improved. The writings are too small and cannot be read, you could enlarge the image to full page or distribute the diagram in height.

155) Do not use abbreviation such as i.e. I have seen that you often use this abbreviation, so I will not repeat this advice again, it also applies to the other occurrences.

166) Introduce adequately the topic (GCN)

174) Introduce adequately the topic (LeakyReLU activation function) or add references to allow the reader to learn more about the topic

260) It seems to me that this is a dataset available online, if it is true it would be advisable to add a link where you can download it so that the reader can reproduce your experiment.

265) It seems to me that this is a dataset available online, if it is true it would be advisable to add a link where you can download it so that the reader can reproduce your experiment.

294) Introduce adequately the topic (Ablation study)

317) Table 2 must be improved. You must specify in the caption of the table what those numbers represent.

317) Table 3 must be improved. You must specify in the caption of the table what those numbers represent.

362) Introduce adequately the topic (confusion matrix)

Reviewer 2 Report

The abstract should be revised. It should highlight the improvements compared to the state-of-the-art methods with numbers or percentages. 

In the introduction, it will be better to enumerate the problems of existing and related existing approaches before stating the proposed solution’s contributions.

Contributions are not clear and in my opinion, this work is roughly the same as already exists in the literature.

It is suggested to add the reference "Detecting Third Umpire Decisions & Automated Scoring System of Cricket" in the CNN description. 

Reviewer 3 Report

The work is devoted to recognition of collective activities of group of persons and individual share of this activity of the person in the group. The main idea is to utilize both the shape of person's body alone (intra-actor) and the relations of same body parts across the group (inter-actor). The method workflow consists of successive feature extraction, intra-actor processing, inter-actor processing and inferring the group activity. All stages are based on neural networks. The intra-actor and inter-actor data are represented as graphs (2D matrices, or 3D matrices for graphs with several type of edges). Numerical experiments are performed with two publicly available databases and compared against existing rivals.

The paper is clearly written, presents all necessary parts to understand the authors' research and contribution. The following are minor required corrections.

1. It seems $\beta$ letter should be somewhere in formula (7).

2. Formula (10) is strange. It seems $j\in N_i$ should be summation index.

3. I could not find whether set of possible activities of one person and set of possible activities of a group should be same sets or can differ in your approach. Please describe.

Round 2

Reviewer 1 Report

The authors addressed all the reviewer's comments with sufficient attention and modified the paper consistently with the suggestions provided. The new version of the paper has improved significantly both in the presentation that is now much more accessible even by a reader not expert in the sector, and in the contents that now appear much more incisive.

Reviewer 2 Report

The authors addressed all of the comments properly